# AngleRoCL: Angle-Robust Concept Learning for Physically View-Invariant T2I Adversarial Patches

**Wenjun Ji**[1,2]    **Yuxiang Fu**[1,2]    **Luyang Ying**[1,2]    **Deng-Ping Fan**[1,2]
**Yuyi Wang**[3]    **Ming-Ming Cheng**[1,2]    **Ivor Tsang**[4]    **Qing Guo**[1,2*]

[1]NKIARI, Shenzhen Futian    [2]VCIP, CS, Nankai University    [3]CRRC Zhuzhou Institute
[4]CFAR and IHPC, Agency for Science, Technology and Research (A*STAR)
wenjj@mail.nankai.edu.cn    tsingqguo@ieee.org

## Abstract

Cutting-edge works have demonstrated that text-to-image (T2I) diffusion models can generate adversarial patches that mislead state-of-the-art object detectors in the physical world, revealing detectors' vulnerabilities and risks. However, these methods neglect the T2I patches' attack effectiveness when observed from different views in the physical world (*i.e.*, angle robustness of the T2I adversarial patches). In this paper, we study the angle robustness of T2I adversarial patches comprehensively, revealing their angle-robust issues, demonstrating that texts affect the angle robustness of generated patches significantly, and task-specific linguistic instructions fail to enhance the angle robustness. Motivated by the studies, we introduce Angle-Robust Concept Learning (AngleRoCL), a simple and flexible approach that learns a generalizable concept (*i.e.*, text embeddings in implementation) representing the capability of generating angle-robust patches. The learned concept can be incorporated into textual prompts and guides T2I models to generate patches with their attack effectiveness inherently resistant to viewpoint variations. Through extensive simulation and physical-world experiments on five SOTA detectors across multiple views, we demonstrate that AngleRoCL significantly enhances the angle robustness of T2I adversarial patches compared to baseline methods. Our patches maintain high attack success rates even under challenging viewing conditions, with over 50% average relative improvement in attack effectiveness across multiple angles. This research advances the understanding of physically angle-robust patches and provides insights into the relationship between textual concepts and physical properties in T2I-generated contents. We released our code in https://github.com/tsingqguo/anglerocl.

## 1    Introduction

Recent advances in deep learning have revealed that text-to-image (T2I) diffusion models can generate adversarial patches capable of misleading state-of-the-art object detectors in physical environments [48, 58]. Unlike traditional optimization-based adversarial patch generation methods [4, 23, 62] that rely on gradient-based pixel manipulations and often struggle to transfer to physical settings and detection models due to environmental factors like printer's color shifting, T2I generation approaches offer significant advantages (*e.g.*, low-cost, model-agnostic, and transferable) in the physical deployment [48, 58]. These patches exploit vulnerabilities in detection systems, creating significant security concerns for critical applications such as autonomous driving and surveillance. However, existing approaches to generating T2I adversarial patches have largely overlooked a crucial real-world challenge: maintaining attack effectiveness when patches are viewed from different angles

---

[*]Corresponding author

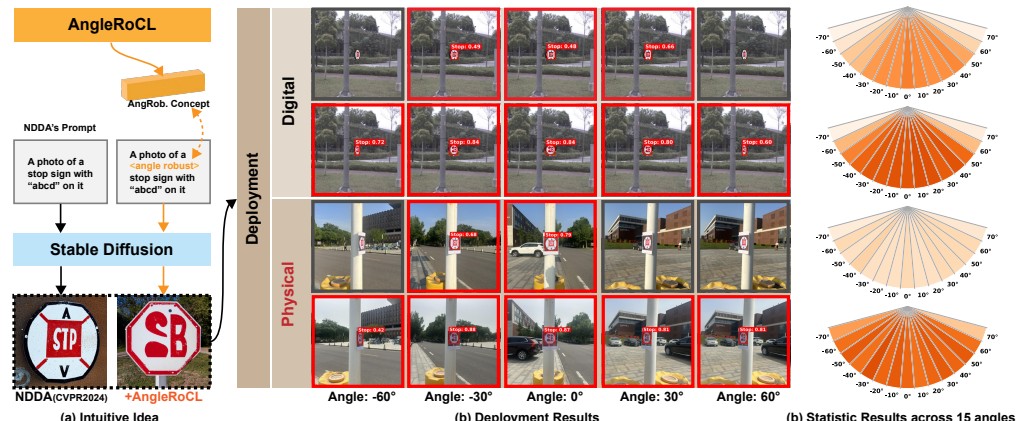

**Figure 1:** (a) Intuitive idea of AngleRoCL: incorporating the learned angle-robust concept into textual prompts to generate patches that maintain attack effectiveness across multiple viewing angles. (b) Digital and physical deployment results AngleRoCL patches (bottom rows) vs. NDDA baseline (top rows). (c) Statistical results across 15 angles, where orange intensity represents attack success rates.

in physical environments. As shown in Fig. 1, NDDA's patches can only mislead the stop sign detector (YOLOv5) at the front angles while failing at other angles in both digital and physical settings.

In practical scenarios, adversarial patches are typically observed from various viewpoints, making angle robustness essential for sustained attack effectiveness. Current approaches predominantly generate patches that demonstrate attack capabilities only under fixed or limited viewing angle ranges, significantly limiting their effectiveness in real-world applications. For example, patches designed to mislead stop sign detection may work when viewed head-on but fail when observed from oblique angles, reducing their practical impact.

In this paper, we first conduct a comprehensive investigation into the angle robustness of T2I adversarial patches. Our study reveals several important findings: ❶ the angle robustness of T2I adversarial patches varies significantly depending on the textual prompts used; ❷ simply augmenting prompts with task-specific linguistic instructions (*e.g.*, "detectable at multiple angles in all directions") fails to enhance angle robustness; and ❸ certain robust feature-related prompt elements have greater influence on angle robustness than others. Please refer to Sec. 3 for details.

Motivated by these insights, we introduce angle-robust concept learning (AngleRoCL), a novel approach that learns a concept representing the capability to generate angle-robust adversarial patches. Unlike previous methods that require specific optimization for each patch/environment, our approach encodes angle robustness as a learned concept in the embedding space of T2I models. The learned concept can then be incorporated into any subject-related textual prompt, guiding diffusion models to generate patches with attack effectiveness that inherently remains robust across multiple viewing angles. AngleRoCL offers several key advantages over existing approaches: ❶ environment-free learning that eliminates the need for user-provided environment images; ❷ detector-guided optimization that leverages feedback from target detectors to supervise the concept learning process; and ❸ consistent attack effectiveness maintained across diverse viewing angles. Through extensive experiments in both digital and physical environments, we demonstrate that AngleRoCL significantly enhances the angle robustness of T2I adversarial patches compared to baseline methods (See Fig. 1 (a)), achieving about relative improvement of 58.96% in digital environments and 82.41% in physical environments in attack effectiveness across multiple viewing angles. This research advances the understanding of physically robust adversarial patches and provides valuable insights into the relationship between textual concepts and physical properties in T2I-generated content.

## 2 Related Works

**Physical adversarial attacks.** Latest research indicates that deep neural network (DNN)-based object detection systems are vulnerable to adversarial patches that can induce erroneous outputs [13, 17, 36, 39, 41, 51, 19, 16, 60, 28, 30]. Physical adversarial patches pose severe threats to critical systems, including autonomous driving [55], classification models [5, 13, 63], and other safety-critical

domains [9, 10, 20, 50, 52]. These patches are highly reproducible and deployable [51], making it essential to investigate their attack effectiveness [5, 52]. Given that these physical adversarial patches are highly reproducible and deployable [48], their attack effectiveness [5, 52, 22, 18], stealthiness [11, 23, 26, 54, 25], and scenario plausibility [6, 58] warrant in-depth investigation. However, achieving efficient physical adversarial attacks remains challenging. Taking physical attacks targeting stop sign as a representative case, existing studies predominantly generate adversarial patches that only demonstrate effective attack capabilities under fixed or narrow viewing angle ranges, thereby exhibiting constrained attack effectiveness in practical applications. To address this limitation, our research focuses on generating angle-robust adversarial patches capable of maintaining attack efficacy across multiple viewing angles in physical deployment scenarios.

**T2I generation for attacks.** Traditional adversarial attacks predominantly operate within the digital domain, conducting targeted manipulations by introducing perturbations to image [7, 17, 39, 40, 41, 51] or by altering pixel values [51]. While these perturbations maintain visual imperceptibility to human observers, their efficacy degrades significantly when deployed in physical environments due to environmental factors [1, 35, 33, 59]. Recent advancements in diffusion models [21, 43, 45, 47] have enabled researchers to employ text-to-image (T2I) models for directly generating adversarial patches [38, 56, 34, 37, 31, 32]. Among them, the Natural Denoising Diffusion (NDD) attack [48, 58] achieves superior attack effectiveness compared to traditional methods in the physical world due to non-robust features that are predictive but incomprehensible to humans.

**Angle robustness enhancements.** The deployment of adversarial patches in practical physical environments necessitates systematic consideration of various complex environmental factors, with detector viewing angle variation constituting a critical influencing parameter. Previous research on angle robustness primarily focused on 3D object camouflage domains, where certain methods attempted to generate or modify surface textures of physical 3D objects to achieve multi-view disguise effects [2, 8, 12, 27, 42, 49, 64]. However, these specially engineered textures suffer from critical limitations including overfitting to specific viewing angles, poor transferability across objects, and visually unnatural appearances [8, 24]. In the 2D domain, existing studies primarily concentrate on evasion attacks - such as attaching adversarial stickers to existing objects to disrupt detector recognition processes [13, 57]. Some approaches have explored transformation-aware optimization frameworks that incorporate diverse geometric and photometric variations to enhance adversarial patch robustness in real-world deployments [5, 3, 49], yet demonstrate inadequate angular robustness [9, 62]. To our knowledge, no prior work has formally conceptualized angular robustness for adversarial patches, nor successfully integrated this property into practical patch generation frameworks for physical deployment scenarios.

## 3 Preliminaries and Discussions

### 3.1 T2I Adversarial Patches

The recent work [48] demonstrates that text-to-image (T2I) models can generate patches capable of misleading object detectors in the physical world by simply modifying input text prompts to remove robust features. This operational pipeline can be formalized as follows: given a textual prompt $\mathcal{T}$ describing both the target subject (*e.g.*, stop sign) and its associated robust features (e.g., shape, color, text, and pattern), a stable diffusion model $\mathcal{M}(\cdot)$ processes $\mathcal{T}$ to generate a patch $\mathbf{P}$, which can be formulated as $\mathbf{P} = \mathcal{M}(\mathcal{T})$. The work demonstrates that by modifying the textual prompt $\mathcal{T}$ to remove robust features, the T2I model generates patches that can effectively mislead various state-of-the-art object detectors in a black-box manner in physical-world scenarios. We denote the patch as *T2I adversarial patches* to distinguish them from conventional adversarial patches.

### 3.2 Empirical Studies on Angle Robustness

**Angle robustness testing.** Although effective, previous works [48, 58] overlook the angle robustness of T2I adversarial patches—their effectiveness when viewed from different angles. To address this gap, we comprehensively evaluate the angle robustness of T2I adversarial patches. Specifically, we selected "stop sign" as our test subject, with the dual objective of generating patches that ❶ appear non-suspicious to human observers and ❷ cause detectors to misclassify them as a stop sign. Following the NDDA method [48], we constructed a prompt set $\{\mathcal{T}_i | i \in [1, ..., 15]\}$ including 15

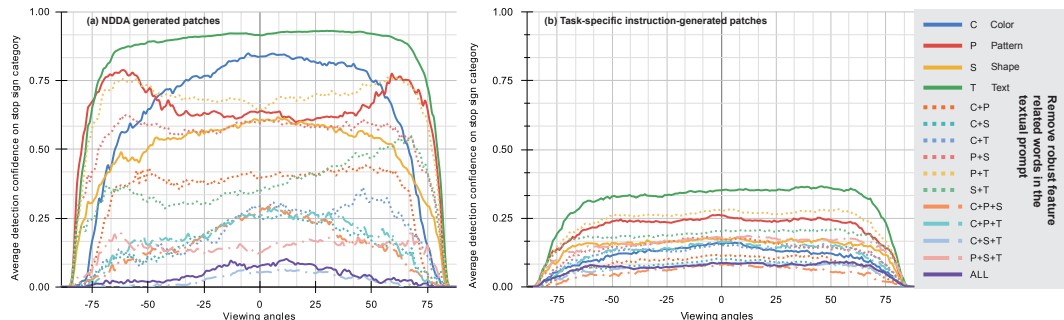

**Figure 2:** Average detection confidence (*i.e.*, $\mathcal{R}_\theta(\mathcal{T}_i)$) at different viewing angles (*i.e.*, $\theta$) across image sets (*i.e.*, $\mathbf{I}_\theta^{(i,j)}$) generated by given attack strategy ($i \in [1, ..., 15], j \in [1, ..., K], \theta \in (-90°, 90°), \nabla\theta = 1°$). The y-axis labels indicate average detection confidence and the x-axis labels indicate different viewing angles. The rightmost panel demonstrates removed features: Color (C), Pattern (P), Shape (S), and Text (T). Patch generated by (a) NDDA prompt and (b) Task-specific instruction prompt.

types of prompt by removing different robust features from benign stop sign descriptions [2].See supp. for more details. For each prompt $\mathcal{T}_i$, we generated $K$ patches $\{\mathbf{P}_i^j | j \in [1, ..., K]\}$. For each patch $\mathbf{P}_i^j$, we inserted it into an environment and captured images $\mathbf{I}_\theta^{(i,j)}$ from various angles $\theta$, spanning the range $\theta \in (-90°, 90°)$ with a sampling interval of $\nabla\theta = 1°$. An angle of zero (*i.e.*, $\theta = 0°$) represents a frontal view (See Fig. 1 (b)). We then fed each image $\mathbf{I}_\theta^{(i,j)}$ into an object detector and recorded the false-positive confidence score $s_\theta^{(i,j)}$ (*i.e.*, probability of misclassification as "stop sign"). This procedure generated 180 test images per patch $\mathbf{P}_i^j$, resulting in $K \times 180$ evaluations per prompt $\mathcal{T}_i$. We defined the angle robustness metric for prompt $\mathcal{T}_i$ at viewing angle $\theta$ as the average of $s_\theta^{(i,j)}$ across $K$ adversarial patches, *i.e.*, $\mathcal{R}_\theta(\mathcal{T}_i) = \frac{1}{K}\sum_{j=1}^{K} s_\theta^{(i,j)}$. We evaluated the 15 prompt types by setting $K = 50$ and digitally inserting the generated patches into environmental images.

We show the visualization of $s_\theta^{(i,j)}$ in Fig. 2 (a), and observe that: ❶ Most prompts $\mathcal{T}_i$ display centripetal confidence increase across 180° viewing angles ; ❷ Attack efficacy roughly presents an inverse correlation with the number of ablated features ; ❸ For $\mathcal{T}_i$ that removed the same counts of features, the performance differences are still significant ; ❹ Only a minority of prompts achieve statistically significant attack success. The experimental results demonstrate significant variance in attack efficacy across different prompts $\mathcal{T}_i$ under varying viewing angles.

**Task-specific instruction for angle robustness enhancement.** To investigate potential improvements, we implemented another attack strategy: augmenting the original prompts in $\{\mathcal{T}_i | i \in [1, ..., 15]\}$ with task-oriented narrative instructions, such as appending the phrase "detectable at multiple angles in all directions". Three modified prompt configurations were engineered: ❶ Prefix-enhanced: $\{\mathcal{T}_p + \mathcal{T}_i\}$. ❷ Infix-integrated: $\{\mathcal{T}_i + \mathcal{T}_m + \mathcal{T}_i\}$. ❸ Suffix-appended: $\{\mathcal{T}_i + \mathcal{T}_s\}$. All other experimental parameters remained unchanged except sample size $K$ ($\theta \in (-90°, 90°), \nabla\theta = 1°$). As illustrated in Fig. 2 (b), the augmented prompts exhibited significant degradation in angular robustness metrics. This empirical evidence suggests that simply incorporating task-specific linguistic instructions fails to enhance the angle robustness. Consequently, we conclude that current T2I text encoders lack the capacity to interpret abstract, goal-oriented narrative commands for angle-robust purposes.

## 4 Methodology: Angle-Robust Concept Learning (AngleRoCL)

Our analyses show different textual prompts significantly affect angle robustness, while simple task-specific linguistic instructions fail to enhance this property. To address this limitation, we propose AngleRoCL to encode angle robustness as a latent concept, which can be plugged into textual prompts and guide T2I to generate patches that maintain attack efficacy across viewpoint variations while preserving physical deployability. Fig. 3 shows our AngleRoCL's pipeline.

---

[2]In NDDA, we have an original prompt for "stop sign", *i.e.*, "a photo of a stop sign". We can remove robust features (*i.e.*, color (C), pattern (P), Shape (S), and Text (T)) by adding constraints into the standard description.

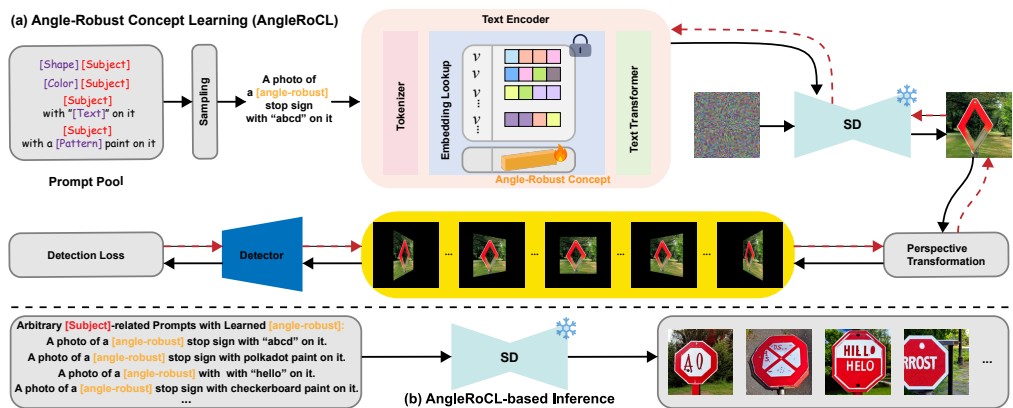

**Figure 3:** (a) Pipeline of our angle-robust concept learning (AngleRoCL). The latent code represents the learned concept `<angle-robust>`. (b) shows the inference results when we use the learned concept.

## 4.1 Problem Formulation

Given a pre-trained stable diffusion model $\mathcal{M}(\cdot)$, we generate an adversarial patch by feeding the model with a texture prompt $\mathcal{T}$ containing the subject and related constraints, such that $\mathbf{P} = \mathcal{M}(\mathcal{T})$. When this generated patch $\mathbf{P}$ is inserted into an environment denoted as E, it produces an image from observation angle $\theta$, that is, $\mathbf{I}_\theta = \text{Obs}(\mathbf{P}, \text{E}, \theta)$. Our objective is to ensure the patch consistently misleads object detectors across various viewing angles. Intuitively, following the existing solution, we can add optimized adversarial noise to the generated patch to achieve the goal [62]. However, such a solution needs to perform the optimization for each patch, and the optimized noise can hardly be transferred into the physical world and keep attack effectiveness against different detectors. In this work, we propose to learn a concept to represent the capability of generating angle-robust adversarial patches, and the learned concept could be defined through new "words" in the embedding space and be inserted into any existing subject-related prompts to generate angle-robust patches. We can formulate the angle robust concept learning (AngleRoCL) as

$$\mathcal{C} = \arg\min_{\mathcal{C}_*} \mathbb{E}_{\theta\sim(-90^\circ, 90^\circ)} \mathcal{L}_{\text{det}}(\text{Obs}(\mathbf{P}, \text{E}, \theta), y)\mathrm{d}\theta, \text{subject to}, \mathbf{P} = \mathcal{M}(\Lambda(\mathcal{T}, \mathcal{C}_*)), \quad (1)$$

where $\mathcal{C}$ denotes the words representing the learned angle-robust concept and $\Lambda(\mathcal{T}, \mathcal{C}_*)$ is the function to insert the concept words into the textual prompt. The loss function $\mathcal{L}_{\text{det}}(\cdot)$ measures whether the attack objective indicated by $y$ has been achieved, for example, the generated patch is misdetected as "Stop Sign". Once we obtain $\mathcal{C}$, we can incorporate it into any subject-related textual prompts to generate patches that maintain robust performance across multiple viewing angles (See Fig. 3 (b)).

## 4.2 AngleRoCL as the Angle-aware Textual Inversion

Textual inversion was proposed for personalizing diffusion-based generation, enabling the model to learn a specific concept of an object or style from user-provided images [15]. This technique allows for consistent regeneration of the same object by simply incorporating the concept's learned token embedding into the generation model's input prompt. Inspired by this approach, we formulate angle-robust concept learning as a textual inversion problem. Unlike previous work, AngleRoCL should have the following key properties: ❶ **Environment-free learning.** Rather than requiring user-provided images of a concept, our learned angle-robust concept remains effective across any environment without relying on specific environmental images. ❷ **Detector-guided optimization.** AngleRoCL should leverage feedback from the target detector to directly supervise the optimization of the angle-robust concept. ❸ **Consistent attack effectiveness across viewing angles.** During the learning process, we optimize not only for image quality but specifically for adversarial performance maintained across multiple angular perspectives. *Our method should uniquely combine angle-awareness with environment-free concept learning through detector feedback across diverse observation angles.*

Specifically, we start with the input textual prompt with the $\Lambda(\mathcal{T}, \mathcal{C})$ where we tend to insert the targeted $\mathcal{C}$ (*i.e.*, "angle-robust") into an existing textual prompt $\mathcal{T}$. Then, we have a text encoder like CLIP to extract the embeddings of $\Lambda(\mathcal{T}, \mathcal{C})$ and get $\mathbf{F}_\mathcal{T}$ and $\mathbb{F}_\mathcal{C}$. Here, we aim to learn a new

embedding of $\mathcal{C}$ to replace the original one (*i.e.*, $\mathbb{F}_\mathcal{C}$), which denotes the angle-robust concept and should make the subsequent patch generation angle-robust. Then, we can reformulate the Eq. (1)

$$\mathbf{F}_\mathcal{C} = \arg\min_{\mathbf{F}_{\mathcal{C}_*}} \mathbb{E}_{\theta \sim (-90°, 90°)} \mathcal{L}_{\det}(\mathrm{Obs}(\mathbf{P}, \mathrm{E}, \theta), y), \text{ subject to, } \mathbf{P} = \mathcal{M}([\mathbf{F}_\mathcal{T}, \mathbf{F}_{\mathcal{C}_*}]). \qquad (2)$$

However, it requires high costs to sample different angles during the optimization. Hence, we perform the projective transformation on the image captured at the angle $\theta = 0$ to produce the images captured at other angles. Then, the Eq. (2) is reformulated as

$$\mathbf{F}_\mathcal{C} = \arg\min_{\mathbf{F}_{\mathcal{C}_*}} \mathbb{E}_{\theta \sim (-90°, 90°)} \mathcal{L}_{\det}(\mathrm{Proj}(\mathbf{I}_{0°}, \theta), y), \qquad (3)$$

$$\text{subject to, } \mathbf{I}_{0°} = \mathrm{Obs}(\mathbf{P}, \mathrm{E}, 0°), \mathbf{P} = \mathcal{M}([\mathbf{F}_\mathcal{T}, \mathbf{F}_{\mathcal{C}_*}]), \qquad (4)$$

where $\mathrm{Proj}(\cdot)$ denotes the projective transformation corresponding to the observation angle.

**Angle robustness loss.** We adopt a loss function $\mathcal{L}_{\det}(\cdot)$ to maximize the adversarial effect across multiple viewing angles, which is defined as follows

$$\mathcal{L}_{\det}(\mathbf{I}_\theta, y) = \max(y - \mathrm{Det}(\mathbf{I}_\theta), 0) \cdot \lambda, \qquad (5)$$

where $\mathrm{Det}(\mathbf{I}_\theta)$ is the confidence score of a detector on the interested category (*e.g.*, "stop sign"). Here, we use YOLOv5. $y$ is the detection threshold and $\lambda$ is a scaling factor. This loss penalizes viewing perspectives where detection confidence falls below the threshold, encouraging wide-angle effectiveness. By minimizing this loss across different angles, we optimize the $\mathbf{F}_\mathcal{C}$ to guide the patch generation that maintains high detection confidence across all viewing angles.

**Angle-robust patch generation.** Once the angle robustness concept is learned and encapsulated in the `<angle-robust>` token, generating angle-robust adversarial patches becomes remarkably straightforward. Our method's key advantage lies in its plug-and-play nature, requiring minimal modifications to existing text-to-image workflows. To generate angle-robust patches, users simply need to load our trained embedding into Stable Diffusion and incorporate the `<angle-robust>` placeholder into their existing attack prompts. For example, a standard NDDA prompt like "a blue square stop sign" can be enhanced to "a `<angle-robust>` blue square stop sign" to produce a patch with inherent angle robustness. This approach is compatible with various Natural Denoising Diffusion (NDD) attack frameworks, including NDDA [48] and MAGIC [58], without requiring any architectural modifications or additional optimization steps.

### 4.3 Implementation Details and Physical Deployment

**Implementation details.** We implement our approach using the Stable Diffusion v1.5 as the base diffusion model with DPMSolver++ for denoising. We set the classifier-free guidance scale to 7.5 and use 25 denoising steps. For the angle-robust concept, we use CLIP embedding of `<angle-robust>` as the initialization. Focusing on the stop sign category, we utilize a total of 39 NDDA prompt templates that incorporate various robustness features including shape (*e.g.*, square, triangle), color (*e.g.*, blue, yellow), text (*e.g.*, "hello", "abcd", "world"), and pattern (checkerboard, polkadot). Examples include "a blue square stop sign", "a stop sign with 'hello' on it", and "a yellow triangle stop sign with polkadot paint on it" (see supp. for more details). During the training process, we sample 9 angles, *i.e.*, $\{-72°, -54°, -36°, -18°, 0°, 18°, 36°, 54°, 72°\}$, which are symmetrically and equally spaced within the range from $-90°$ to $90°$ (excluding the endpoints). The detection loss parameter $y$ and the scaling factor $\lambda$ are respectively set to 0.8 and 10. We train for 50,000 steps using AdamW optimizer with learning rate $10^{-4}$, updating only the `<angle-robust>` embedding while keeping all other parameters frozen.

**Physical deployment.** The physical deployment of our angle-robust patches follows a straightforward process validating their real-world effectiveness. We directly print the generated adversarial patches using a standard color printer on regular office paper, without requiring specialized printing techniques or materials. These patches are then affixed to target objects (stop signs) in various environments.

## 5 Experimental Results

### 5.1 Setups

**Digital & Physical environments.** Following the evaluation protocol in [58], we adopt the nuImage dataset [14], selecting one representative image from each of the six car-mounted camera views (front,

front left, front right, back, back left, back right) for digital evaluation. We also validate AngleRoCL in real-world scenarios, which are collected by ourselves. The main paper presents results from one physical environment, while cross-scene validation across three physical environments is provided in the supplementary material. See supp. for more details.

**Baseline methods.** To demonstrate the effectiveness of AngleRoCL in improving angle robustness for physical adversarial patches, we establish comparisons with four baseline methods: one traditional physical adversarial patch approach (AdvPatch [5]) and two NDD attacking methods (NDDA [48] and MAGIC [58]). ❶ For the traditional methods, we generate AdvPatch with Adversarial Robustness Toolbox , selecting only the highest-performing patch from each method for evaluation. ❷ For NDD attacking methods, we follow the methodology outlined in previous work [58], generating 50 patches for each "remove text or pattern" text prompt, and randomly selecting 100 patches as our NDDA baseline set. ❸ For MAGIC, we utilize its generation component to produce 100 patches for each digital environment using identical prompt configurations to ensure fair comparison. To isolate the effect of our proposed approach, NDDA+AngleRoCL and MAGIC+AngleRoCL patches were generated concurrently with the same seed and generator. The only difference in the generation process was the inclusion of our angle-robust embedding in the text prompt. We extend our evaluation from digital to physical environments, acknowledging the higher cost and complexity of physical experimentation. For physical validation, we selected 25 matched pairs of patches from each method (NDDA, NDDA+AngleRoCL, MAGIC, and MAGIC+AngleRoCL).

**Generator & Detectors.** For consistency with prior work [48, 58], we employ Stable Diffusion v1.5 [46] as our image generator. Our evaluation spans multiple object detection architectures, including YOLOv5 [29], YOLOv3 [44], Faster R-CNN [44], and RT-DETR [61]. We additionally evaluate against YOLOv10 [53], a more robust contemporary detector, as our primary benchmark. For traditional adversarial patches, we use the same detectors for both training and evaluation as in their original works to ensure fair comparison. For implementation frameworks, YOLOv5 and YOLOv10 are evaluated using the API from ultralytics, while Faster R-CNN, YOLOv3, and RT-DETR are implemented through the MMDetection framework to ensure consistent evaluation procedures.

**Evaluation metrics.** Previous studies have primarily relied on **Attack Success Rate (ASR)** to evaluate adversarial patch effectiveness. However, this conventional metric typically assesses performance only under fixed or limited viewpoints, failing to capture robustness across diverse viewing angles—a critical factor for real-world deployment. To address this limitation, we introduce a novel evaluation metric: **Angle-Aware Attack Success Rate (AASR)**. This metric comprehensively quantifies a patch's effectiveness across varied viewing perspectives. Let $\Omega$ represent the complete angular space of interest. The AASR is defined as a weighted integral of ASR across this angular domain: $\text{AASR} = \int_{\Omega} w(\theta) \cdot \text{ASR}(\theta) \cdot d\theta \times 100\%$ where $\text{ASR}(\theta)$ represents the traditional attack success rate when patches are viewed at angle $\theta \in \Omega$, and $w(\theta)$ is a normalized weighting function such that $\int_{\Omega} w(\theta) \cdot d\theta = 1$. In our evaluation, we adopt uniform weighting, *i.e.*, $w(\theta) = \frac{1}{|\Omega|}$ for all angles, ensuring equal contribution from each viewing angle to the AASR calculation. This generalized formulation allows for evaluation across arbitrary angular domains, accommodating various real-world deployment scenarios.

## 5.2  Digital Comparative Results

We evaluate AngleRoCL in digital environments across multiple detectors, comparing 4 T2I-based attacks (NDDA, NDDA+AngleRoCL, MAGIC, MAGIC+AngleRoCL) with a traditional physical adversarial approach (AdvPatch). Our evaluation simulates angles from $-90°$ to $90°$ using projective transformations, with patches placed at environment centers to eliminate positional bias. Table 1 summarizes the AASR performance across 5 detectors and 6 environments. We have the following observations: ❶ AngleRoCL significantly outperforms traditional physical adversarial approach. While AdvPatch only achieves 5.54% average AASR with substantial fluctuations across environments (0.00% to 14.21%), our NDDA+AngleRoCL reaches 36.02% and MAGIC+AngleRoCL 32.51%. ❷ AngleRoCL consistently enhances angle robustness of NDD-based methods, increasing NDDA's average AASR from 23.79% to 36.02% (51.4% improvement) and MAGIC's from 26.26% to 32.51% (23.8% improvement). ❸ Improvements persist across diverse environments, from 47.61% AASR in favorable Environment ② to 24.24% in challenging Environment ④. ❹ AngleRoCL enhances performance across all detection architectures, including YOLOv10 (72.8% improvement) and YOLOv5 (81.2% improvement), validating its cross-detector transferability.

**Table 1:** Angle-Aware Attack Success Rate (AASR) in digital environments. Results across five detectors in six environments, measured from -90° to 90° with 1° intervals. Best average highlighted in red, second best in blue. Best detector results **underlined+bold**, second best **bold**.

| Environment | Method | Faster R-CNN | YOLOv3 | YOLOv5 | RT-DETR | YOLOv10 | Avg. |
|---|---|---|---|---|---|---|---|
| Environment ① | AdvPatch | 11.51% | 2.52% | 0.00% | 0.00% | 0.00% | 0.78% |
| | NDDA | 25.96% | 1.88% | 14.62% | 14.57% | 10.02% | 13.41% |
| | NDDA+AngleRoCL | **39.79%** | **6.38%** | **36.58%** | 19.30% | **23.82%** | 25.17% |
| | MAGIC | 29.35% | 1.97% | 18.55% | **25.52%** | 11.67% | 17.41% |
| | MAGIC+AngleRoCL | **39.72%** | 5.94% | 34.08% | **30.07%** | 20.24% | 26.01% |
| Environment ② | AdvPatch | 23.31% | 43.53% | 2.25% | 1.97% | 0.00% | 14.21% |
| | NDDA | 41.64% | 32.86% | 29.99% | 45.43% | 28.99% | 35.78% |
| | NDDA+AngleRoCL | **50.67%** | **50.76%** | **46.04%** | **47.64%** | **42.95%** | 47.61% |
| | MAGIC | 44.22% | 34.94% | 30.76% | 42.80% | 31.12% | 36.77% |
| | MAGIC+AngleRoCL | **45.45%** | **43.34%** | **42.32%** | **44.29%** | **39.98%** | 43.08% |
| Environment ③ | AdvPatch | 8.99% | 15.73% | 8.71% | 0.00% | 0.00% | 6.69% |
| | NDDA | 30.27% | 10.60% | 25.67% | 23.81% | 22.18% | 22.51% |
| | NDDA+AngleRoCL | **43.94%** | **25.74%** | **43.38%** | **33.40%** | **36.11%** | 36.51% |
| | MAGIC | 31.13% | 14.01% | 32.46% | 25.29% | 24.70% | 25.52% |
| | MAGIC+AngleRoCL | **36.04%** | **25.99%** | **42.71%** | **31.11%** | **34.16%** | 34.00% |
| Environment ④ | AdvPatch | 0.00% | 0.00% | 0.00% | 0.00% | 0.00% | 0.00% |
| | NDDA | 20.80% | 3.61% | 10.31% | 18.07% | 13.11% | 13.18% |
| | NDDA+AngleRoCL | **31.25%** | **6.74%** | **27.34%** | 29.99% | **25.87%** | 24.24% |
| | MAGIC | 21.24% | 3.71% | 14.46% | 22.90% | 15.98% | 15.66% |
| | MAGIC+AngleRoCL | **28.42%** | **6.75%** | **22.68%** | **31.03%** | 22.24% | 22.22% |
| Environment ⑤ | AdvPatch | 9.55% | 10.11% | 0.00% | 6.46% | 0.00% | 5.22% |
| | NDDA | 38.70% | 12.41% | 29.07% | 52.45% | 21.93% | 30.91% |
| | NDDA+AngleRoCL | **47.52%** | **26.26%** | **42.80%** | **54.23%** | **36.50%** | 41.46% |
| | MAGIC | 38.04% | 14.67% | 34.19% | 53.77% | 26.88% | 33.51% |
| | MAGIC+AngleRoCL | **41.44%** | **24.38%** | **38.74%** | 53.28% | **31.76%** | 37.92% |
| Environment ⑥ | AdvPatch | 0.56% | 31.15% | 0.00% | 0.00% | 0.00% | 6.34% |
| | NDDA | 28.55% | 25.95% | 20.17% | **39.56%** | 20.46% | 26.94% |
| | NDDA+AngleRoCL | **42.02%** | **43.05%** | **39.14%** | **45.13%** | **36.41%** | 41.15% |
| | MAGIC | **30.53%** | 28.56% | 24.40% | 35.49% | 24.58% | 28.71% |
| | MAGIC+AngleRoCL | 29.72% | **33.24%** | **31.81%** | 37.16% | **27.33%** | 31.85% |

**Table 2:** Angle-Aware Attack Success Rate (AASR) in physical environment. Results across five detectors, measured from -70° to 70° with 10° intervals. Best average highlighted in red, second best in blue. Best detector results **underlined+bold**, second best **bold**.

| Environment | Method | Faster R-CNN | YOLOv3 | YOLOv5 | RT-DETR | YOLOv10 | Avg. |
|---|---|---|---|---|---|---|---|
| Environment ⑦ | AdvPatch | 0.00% | 0.00% | 0.00% | 0.00% | 0.00% | 0.00% |
| | NDDA | 40.27% | 7.73% | 15.20% | 56.27% | 22.40% | 28.37% |
| | NDDA+AngleRoCL | **69.87%** | **16.00%** | **60.80%** | **69.87%** | **42.20%** | 51.75% |
| | MAGIC | 39.73% | 4.27% | 17.33% | 42.67% | 9.60% | 22.72% |
| | MAGIC+AngleRoCL | **83.73%** | **27.73%** | **72.53%** | **89.60%** | **55.73%** | 65.86% |

## 5.3 Physical Comparative Results

To validate our findings in real-world scenarios, we conducted physical experiments comparing the same six methods as in the digital environment. We printed patches on standard paper and deployed them in a regular road next to a college. Due to the higher costs of physical experiments, we employed a reduced angular sampling strategy compared to the digital evaluation. Observations were made at a fixed distance across 15 viewing angles from −70° to 70° at 10° intervals. Table 2 summarizes the AASR performance across five detectors. The evaluation reveals two findings: ❶ AngleRoCL significantly outperforms traditional approach in physical environments. While AdvPatch completely failed in physical settings (0.00% AASR), our methods demonstrated robust performance with NDDA+AngleRoCL achieving 51.75% and MAGIC+AngleRoCL reaching 65.86% AASR. This stark contrast highlights the superior physical-world transferability of our approach compared to traditional methods that are highly sensitive to environmental changes. ❷ AngleRoCL consistently enhances NDD methods in physical settings, with NDDA+AngleRoCL achieving 51.75% AASR compared to 28.37% for vanilla NDDA (82.4% improvement), and MAGIC+AngleRoCL reaching 65.86% versus 22.72% for original MAGIC (189.9% improvement). These substantial enhancements were consistent across all viewing angles, confirming that our learned angular robustness concept transfers effectively to real-world scenarios without environment-specific optimization.

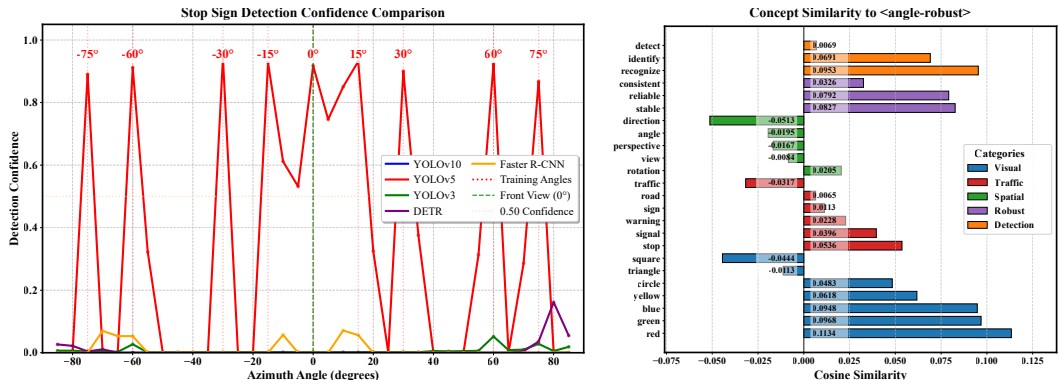

**Figure 4:** (a) Direct optimization shows severe overfitting to training angles (orange) and trained detector (YOLOv5). (b) Cosine similarity between learned `<angle-robust>` embedding and top-correlated tokens.

## 6 Ablation Study and Discussion

**Ablation study.** To validate our approach, we compared AngleRoCL with a direct optimization method that applies gradients directly to patch pixels instead of performing AngleRoCL. In Fig. 4, while directly optimized patches achieved high attack rates at trained angles and on trained detectors, they performed poorly on unseen angles and detectors. In contrast, AngleRoCL maintained consistent performance across all conditions, confirming that our embedding-based concept learning enables critical cross-angle and cross-detector generalization for real-world applications.

**Embedding analysis.** In Fig. 4, the cosine similarity analysis shows our `<angle-robust>` concept has developed meaningful associations with robustness-related features. Color tokens show the highest correlations (red: 0.1134, green: 0.0968, blue: 0.0948), while shapes show interesting patterns with circle being positive (0.0483) and square/triangle negative. To further validate these embedding insights, we regenerated the NDDA dataset using our learned angle-robust concept. Following the original NDDA methodology, we generated patches for all 39 prompts (50 patches per prompt) with and without the `<angle-robust>` token, and evaluated their angle robustness by digitally placing them in the center of blank environments across multiple viewing angles. The results are presented in Table 3. The embedding analysis results align with the experimental results. For NDDA on YOLOv5, removing color features caused the most significant performance drop (from 56.64% to 21.17%), matching our embedding analysis which identified color tokens as having the highest correlation with our concept. Moreover, AngleRoCL maintains a 62.33% detection rate even without color features—a 41.16 percentage point improvement over NDDA, confirming our approach effectively compensates for the removal of critical robustness features.

The correlations clearly show our learned concept captures essential visual attributes proportional to their importance for achieving angle robustness. This explains our method's effectiveness and further suggests angle robustness is fundamentally related to visual elements that maintain distinctive properties across different viewpoints.

**Table 3:** Angle-Aware Attack Success Rate (%) comparison between NDDA and AngleRoCL when robust features are removed. Each row indicates which features were removed from the prompt (checkmark means removed). For each detector and configuration, the best performance is highlighted in **bold**.

| | Removed Robust Features | | | | Object Detectors | | | | | |
|---|---|---|---|---|---|---|---|---|---|---|
| | Shape | Color | Text | Pattern | Faster R-CNN | YOLOv3 | YOLOv5 | RT-DETR | YOLOv10 | Avg. |
| **NDDA** | | | | | 59.58% | 57.98% | 56.64% | 74.96% | 52.57% | 60.35% |
| | ✔ | | | | 30.38% | 28.0% | 27.78% | 44.50% | 23.52% | 30.84% |
| | | ✔ | | | 61.53% | 49.24% | 21.17% | **71.75%** | 20.84% | 44.91% |
| | | | ✔ | | 51.39% | 53.69% | 43.84% | 60.30% | 39.04% | 49.65% |
| | | | | ✔ | 41.95% | 41.26% | 29.69% | 47.94% | 27.98% | 37.76% |
| | ✔ | ✔ | ✔ | ✔ | 38.83% | 31.09% | 6.86% | 52.43% | 9.44% | 27.73% |
| **AngleRoCL** | | | | | **74.16%** | **77.16%** | **77.07%** | **79.61%** | **72.96%** | **76.19%** |
| | ✔ | | | | **37.09%** | **44.11%** | **40.75%** | **54.40%** | **38.69%** | **43.01%** |
| | | ✔ | | | **64.08%** | **66.89%** | **62.33%** | 71.54% | **60.46%** | **65.06%** |
| | | | ✔ | | **53.50%** | **62.31%** | **66.33%** | **69.68%** | **60.33%** | **62.43%** |
| | | | | ✔ | **51.14%** | **62.47%** | **58.38%** | **56.81%** | **54.74%** | **56.71%** |
| | ✔ | ✔ | ✔ | ✔ | **41.60%** | **46.12%** | **44.50%** | **57.01%** | **40.70%** | **45.99%** |

**Cross-Detector generalization** While AngleRoCL improves angle robustness across multiple detectors, the degree varies between architectures. As shown in Table 3, different detectors show varying sensitivities to specific robustness features—when color features are removed, YOLOv5 shows dramatic improvement with our method (21.17% to 62.33%), while RT-DETR shows minimal change (71.75% to 71.54%). Since our framework uses only YOLOv5 for feedback during training, the resulting concept inherently captures YOLOv5's definition of robustness, explaining the most pronounced improvements on YOLOv5 (81.2% relative improvement). This detector-specific bias, while still enabling cross-detector generalization, limits achieving optimal universal robustness. **Limitations.** We trained with only 9 sampled angles in the horizontal plane, which simplifies but doesn't fully capture continuous 3D angle variations, including vertical perspectives in real-world scenarios. While experiments show AngleRoCL significantly enhances patch robustness even with this limited sampling, the optimal angle sampling density and distribution remain unexplored.

## 7 Conclusion

In this paper, we introduced Angle-Robust Concept Learning (AngleRoCL), addressing the critical challenge of maintaining attack effectiveness across multiple viewing angles for T2I adversarial patches. Our comprehensive experiments in both digital and physical environments demonstrate that AngleRoCL significantly outperforms baseline methods without requiring environmental optimization. By encoding angle robustness as a learned concept, our method enables the generation of physically robust adversarial patches with consistent performance across viewpoints. The plug-and-play nature of our approach allows seamless integration with existing T2I attack frameworks. Beyond technical contributions, this work advances understanding of textual concepts and physical properties in diffusion-generated content, providing valuable insights for developing more robust defense mechanisms against angle-invariant adversarial attacks in real-world environments.

## Acknowledgments

This research was supported by the NSFC (No. 62476143), Shenzhen Science and Technology Program (No. JCYJ20240813114237048), "Science and Technology Yongjiang 2035" key technology breakthrough plan project (No. 2025Z053), and Chinese government-guided local science and technology development fund projects (scientific and technological achievement transfer and transformation projects) (No. 254Z0102G). This research is supported by the National Research Foundation, Singapore under its AI Singapore Programme (AISG Award No: AISG4-GC-2023-008-1B), and National Research Foundation, Singapore and Infocomm Media Development Authority under its Trust Tech Funding Initiative. Any opinions, findings and conclusions or recommendations expressed in this material are those of the author(s) and do not reflect the views of National Research Foundation, Singapore, Cyber Security Agency of Singapore, and Infocomm Media Development Authority.

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
