# OpenReview forum: "AngleRoCL: Angle-Robust Concept Learning for Physically View-Invariant Adversarial Patches"
_NeurIPS.cc/2025/Conference — NeurIPS 2025 poster_

### Official Review · Reviewer_uf9C · 2025-06-25

**Clarity:** 2
**Significance:** 3
**Originality:** 3
**Rating:** 4
**Confidence:** 4

**Summary:**

The author propose a new adversarial attack (generating adversarial patches) using textual inversion to improve the efficacy of adversarial patches across different viewpoints. Specifically, the authors propose a method to guide the embedding of their chosen prompt concept <angle-robust> towards regions that result in more effective adversarial patches (generated by a prompted T2I model). Through experiments on various object detection methods, the authors demonstrate the effectiveness of their method compared to other adversarial patch attacks.

**Questions:**

Questions are included in the strengths and weaknesses section above.

**Ethical Concerns:**

["NO or VERY MINOR ethics concerns only"]

**Final Justification:**

The authors have addressed my concerns and questions, which has increased by confidence in my rating. However, the recent results of different type objects introduced new concerns of its generalisability. I believe the study would be improved with more diverse experiments with different objects also.

**Limitations:**

The authors highlight that the largest limitation of their proposed method is due to only sampling 9 viewpoint degrees during training.

There is a lack of discussion on the limitations of this method, or further research.

The paper proposed a novel attack that creates more potent adversarial patches, that not only work within the digital realm but also physical. However, there is little to no discussion on potential defence against this attack. I would encourage a more in-depth discussion about this within their broader impact statement.

**Quality:**

2

**Strengths And Weaknesses:**

Strengths:
- The paper proposed a novel attack method, which is an interesting application of textual inversion for T2I models.
- The authors provide useful figures to help describe their method and its results.
- The authors conduct in expansive companrisons on several different object detection methods and adversarial patch attacks.

Weaknesses:
* At times the writing of the paper makes it difficult to follow:
- Line 60 missing closing bracket *See Fig 1 (a)”
- Lines 67-69: This sentence is not concise, the use of autonomous driving, classification and other domains seem random. I suggest that you replace the
- Lines 125-141 are too short, making the text difficult to read, I would suggest either making Figure 2 smaller or placing it at the top of the page, covering the width of the page.
- Lines 146-147 (point 3), the writing of this point is confusing, please re-write this objective.
- Lines 187-188: “Concept embedding words” is unclear
- Line 198, ‘we start with the input textual prompt with the A(T, C)’

* Within preliminaries, the authors spend most of the text discussing the problem and not the result of the study in Figure 2.
* The paper would benefit from pseudo-code of the proposed method. The authors well describe the problem definition but only mention the the optimiser and learning rate. It would be useful to see some about the whole training process as an algorithm as well as their figure.

* The experimental setup is largely focused on generating stop-signs, the paper would benefit from an explicit definition of the type of target objects they are generating in the experimental setup section.
* The generation of stop signs is a core direction of this paper, it would be good to see expansion to other target objects.

---

> ### Author Rebuttal · Authors · 2025-07-31
>
> ### 1. Attack results on other target objects
>
> **Response**:
>
> Thank you for the insightful question. To address the concern, we conducted additional experiments on two other target objects to demonstrate the broader applicability of our method.
>
> **Experimental Design:** Following NDDA's evaluation scope, we selected fire-hydrant and horse as additional target objects to evaluate object-specific training capabilities. These objects represent different categories—urban infrastructure (fire-hydrant) and natural subjects (horse)—allowing us to assess our method's versatility across diverse visual domains.
>
> **Object-Specific Training Experiment:** We trained dedicated angle-robust concepts for the two target objects, respectively, using their respective NDDA prompt templates. Then, we embed the learned angle-robust concepts into the NDDA prompt to generate angle-robust adversarial patches. **Table R-3** presents the comparative AASR results with/without AngleRoCL:
>
> **Table R-3: Object-Specific Angle-Robust Concept Evaluation**
>
> |Target Object|Method|Faster R-CNN|YOLOv3|YOLOv5|RT-DETR|YOLOv10|Avg.|
> |:-:|:-:|:-:|:-:|:-:|:-:|:-:|:-:|
> |Fire-hydrant|NDDA|16.78%|3.63%|5.50%|23.39%|11.51%|**12.16%**|
> ||NDDA+AngleRoCL|23.95%|6.62%|12.89%|27.14%|19.84%|**18.09%**|
> |Horse|NDDA|19.91%|12.59%|10.32%|26.74%|13.58%|**16.63%**|
> ||NDDA+AngleRoCL|31.86%|23.34%|21.94%|44.70%|24.64%|**29.30%**|
>
> _Note: All experiments in Table R-3 were conducted in Digital Environment 1. For each NDDA prompt, we generated 5 patches, and the reported results represent the average performance across all generated patches._
>
> The results demonstrate that our framework successfully generates effective angle-robust concepts across different target objects, achieving consistent improvements in all categories. This validates that the angle robustness challenge is not unique to stop signs but represents a broader limitation in existing T2I adversarial patch generation methods.
>
>
> ### 2. Pseudo-code and algorithmic details
>
> **Response**:
>
> Thank you for this valuable suggestion. We present the complete pseudo-code for our AngleRoCL training process as follows:
>
> The training process follows a standard textual inversion framework but incorporates angle-aware supervision. For each training iteration, we generate adversarial patches using the learned concept embedding, apply perspective transformations to simulate multi-angle observations (lines 1-20), and optimize the embedding based on detection feedback across all angles (lines 14-16). The key advantage is that only the concept embedding is updated while preserving all original model parameters, ensuring stability and enabling the learned concept to generalize across different prompts and environments.
>
> **Algorithm: Angle-Robust Concept Learning (AngleRoCL)**
>
> ```
> Input:
>   - Pre-trained Stable Diffusion model M (text encoder, UNet, VAE)
>   - NDDA prompt templates P = {p_1, p_2, ..., p_N}
>   - Training angles Θ_train = {θ_1, θ_2, ..., θ_V}
>   - Placeholder token <angle-robust>, initializer token "robust"
>   - YOLO detector D, perspective transformer T
>   - Detection threshold τ, scaling factor λ
>   - Repeat factor R for data augmentation
>
> Initialization:
> 1. Add <angle-robust> token to tokenizer vocabulary
> 2. Initialize embedding E_<angle-robust> ← E_robust (copy initializer embedding)
> 3. Freeze all model parameters except E_<angle-robust>
> 4. Create training dataset: D_train = {(p_i, r) | p_i ∈ P, r ∈ [1,R]}
>
> Training Loop:
> for epoch = 1 to max_epochs do:
>     for each batch in DataLoader(D_train) do:
>         // Text encoding with learned concept
>         1. angle-robust_embed ← TextEncoder(<angle-robust>)
>
>         // Diffusion-based patch generation
>         2. Initialize random latents z_0 ~ N(0,I)
>         3. z_0 ← z_0 * scheduler.init_noise_sigma
>         4. for timestep t in scheduler.timesteps do:
>         5.     z_input ← concat([z_0, z_0])  // For classifier-free guidance
>         6.     z_input ← scheduler.scale_model_input(z_input, t)
>         7.    ε_pred ← UNet(z_input, t, angle-robust_embed)
>         8.    ε_uncond, ε_text ← chunk(ε_pred, 2)
>         9.    ε ← ε_uncond + γ * (ε_text - ε_uncond)  // CFG with scale γ
>         10.    z_0 ← scheduler.step(ε, t, z_0).prev_sample
>         11. end for
>         12. P ← VAE.decode(z_0 / 0.18215)  // Generated adversarial patch
>
>         // Batch multi-angle transformation
>         13. transformed_views ← T(P, Θ_train)  // Transform to all V angles simultaneously
>
>         // Detection and loss computation
>         14. detection_output ← D(transformed_views)
>         15. confidences ← ExtractMaxProb(detection_output)
>         16. loss ← mean(max(τ - confidences, 0)) * λ
>
>         // Backpropagation
>         17. loss.backward()
>         18. optimizer.step()  // Update only E_<angle-robust>
>         19. optimizer.zero_grad()
>
>         // Preserve original embeddings
>         20. E_other ← freeze(E_original[other_tokens])
>     end for
> end for
>
> Output: Learned embedding E_<angle-robust>
> ```
>
> ### 3. Discussion on potential defense solutions
>
> **Response**:
>
> Thank you for this valuable suggestion. Indeed, it is meaningful to discuss potential defense solutions in the broader impact statement. Based on previous works and our preliminary studies, we identify the following defense approaches: (1) Adversarial training methods that fine-tune detectors using our generated adversarial patches. However, this solution may reduce detection accuracy on other object categories and introduce new vulnerabilities. (2) Data purification methods that pre-process input images to disable adversarial patches by removing adversarial patterns. While effective for noise-like adversarial examples, our preliminary results show limited effectiveness against our patches. (3) Adversarial patch detection models that identify and crop patches to avoid their influence. However, this approach requires the detection model to be computationally efficient and demonstrate high generalization across unknown adversarial patches, which necessitates further careful study. We will incorporate and expand upon these defense discussions in our broader impact statement to provide a more comprehensive treatment of both the risks and potential mitigation strategies associated with our work.
>
> ### 4. Writing quality and formatting issues
>
> **Concern**: Various writing issues, including missing closing bracket on Line 60, unclear sentences on Lines 67-69, short paragraphs on Lines 125-141, making text difficult to read, and confusing point descriptions.
>
> **Response**:
>
> We appreciate the reviewers' detailed feedback on specific issues, including missing brackets, unclear sentences, and fragmented paragraphs. We will thoroughly polish the manuscript to address these writing issues and ensure a consistent, clear presentation throughout the submission to avoid similar situations in the revised version.
>
> For example, we will revise the unclear sentences (Lines 67-69) as:
>
> Physical adversarial patches pose severe threats to critical systems, including autonomous driving [42], classification models [4, 12, 47], and other safety-critical domains [8, 9, 16, 36, 38]. These patches are highly reproducible and deployable [34], making it essential to investigate their attack effectiveness [4, 38], ....
>
> ### 5. More limitations discussion
>
> **Response**:
> Thank you for the suggestions. In our original submission, we discussed the limitations on line 367 to 371. To further enhance this part, we have the following revision:
>
> **Limitations.** Although our method significantly enhances the angle robustness of generated patches, the current version has several limitations that warrant further investigation. (1) Our training is constrained to only 9 sampled angles in the horizontal plane, which simplifies the optimization process but does not fully capture continuous 3D angle variations, including vertical perspectives commonly encountered in real-world scenarios. (2) The current version employs only YOLOv5 for the objective function, which may limit performance improvements on other detection architectures. Our preliminary exploration indicates that combining different detectors during concept training fails to achieve higher angle robustness, suggesting that a deeper study of multi-detector optimization strategies is required. (3) The learned concept presents low transferability across distinct object categories, indicating that developing methods with higher cross-category transferability represents an important direction for future research.

---

> > ### Comment · Reviewer_uf9C · 2025-08-04
> > **Thank-You**
> >
> > I thank the authors for their response to my questions and concerns.
> >
> > Overall I think the authors response has improved the quality of this submission. However, despite the method improving the attack success for horse and hire-hydrant the low attack success rates has caused a new concern about its performance. Specifically, the averages of ~18% and ~29% show smaller improvements compared to the stop-sign results shown in the original submission. Can the authors provide some information/hypothesis on this? Are the attacked models generally more robust outside of stop-signs, or was the method original developed based on analysis of stop signs, which is now impacting its generalisability?

---

> > > ### Author Response · Authors · 2025-08-04
> > >
> > > Dear Reviewer uf9C,
> > >
> > > Thank you for the critical feedback. We would like to provide detailed information and hypothesis to analyze the underlying factors.
> > >
> > > **Performance Difference Analysis:**
> > >
> > > As mentioned in our rebuttal, we conducted generalization experiments by training dedicated angle-robust concepts for fire-hydrant and horse using their respective NDDA prompt templates, while keeping all other framework components unchanged. Upon deeper analysis, we identified several key factors contributing to the performance differences:
> > >
> > > **1. Prompt Template Complexity and Scale:** The NDDA prompt templates vary significantly in complexity across target objects:
> > >
> > > - **Stop-sign**: 4 attribute categories (shape, color, pattern, text) with 15 combination types totaling 39 prompts
> > > - **Fire-hydrant**: 3 attribute categories (shape, color, pattern) with 8 combination types totaling 18 prompts
> > > - **Horse**: 3 attribute categories (shape, color, pattern) with 8 combination types totaling 16 prompts
> > >
> > > The reduced prompt diversity and scale for fire-hydrant and horse may lead to insufficient concept learning.
> > >
> > > **2. Inherent Detector Robustness Variation:** Modern object detectors may exhibit varying robustness levels across different object categories, as stop-signs have more standardized geometric properties compared to the natural complexity of horses or structural variations in fire-hydrants.
> > >
> > > Despite these challenges, our method still achieves substantial relative improvements: fire-hydrant shows 48.8% improvement (12.16% → 18.09%) and horse demonstrates 76.2% improvement (16.63% → 29.30%). These results validate that our approach consistently enhances angle robustness across diverse object categories, confirming the generalizability of our method while highlighting important areas for optimization.  We appreciate the critical feedback, inspiring us to continue to investigate these factors and develop new methods for broader object categories in future work.
> > >
> > > We hope the above analysis alleviates your concerns. Thank you.
> > >
> > > Best,
> > >
> > > Authors

---

> > > > ### Comment · Reviewer_uf9C · 2025-08-05
> > > > **Thank You**
> > > >
> > > > I thank the authors for their prompt response to clarify the differing performances of used objects. Despite the reporting that fire-hydrant adversarial object is still improved, I believe the relative value of 48.8% improvement is a little misleading especially when the original adversarial accuracy is low. I would urge the authors in the final revision to discuss both relative and absolute performance gains of the proposed method.
> > > >
> > > > Considering both the authors response and these new results, I will keep my original rating but I will increase my confidence in the borderline acceptance of this work.
> > > >
> > > > I would like to thank the authors again for their response to my questions and concerns.

---

> > > > > ### Author Response · Authors · 2025-08-05
> > > > >
> > > > > Dear Reviewer uf9C,
> > > > >
> > > > > Thank you for your constructive feedback and increased confidence in our work. We appreciate your suggestions regarding the presentation of relative and absolute performance gains.
> > > > >
> > > > > In our revised version, we will prioritize absolute performance gains as the primary discussion metrics while using relative performance gains as supplementary information to highlight the differences across various target objects. This approach will provide readers with a more comprehensive and balanced understanding of our method's effectiveness and limitations across diverse object categories.
> > > > >
> > > > > We greatly appreciate your thorough review and helpful suggestions during this process.
> > > > >
> > > > > Best regards,
> > > > >
> > > > > Authors

---

### Official Review · Reviewer_he3u · 2025-07-01

**Clarity:** 2
**Significance:** 3
**Originality:** 3
**Rating:** 3
**Confidence:** 4

**Summary:**

The authors propose a solution for generating adversial patches that deceive pretrained models regarless of the angle of view considered for presenting the generated patch. Here the aim is to generate images that are completely different from the expected class while making the model predict the expected class. They focus on "stop_sign" class. They employ a T2I distilation model that is trained to generate angle robust adversial patches in presence of a "robust_angle" token. The training is completed in presence of projections of the adversial patch over the entire range of horizontal views : -90° to 90° with a step of 1°. Specific loss functions are defined for finding the suitable latent representation for the token "robust_angle" so that the generated projected patches always yield the "stop_sign" class.

**Questions:**

Figures 1 and 2 are too small (very hard to read even with zoom and googles) with regard to all the information contained and are poorly commented in text. The are also elements in text, that are not at all visible on the figures (like for instance the % mentioned in line 60).

Sometimes the term patch is a bit ambigous. The patch is the region that is supposed to contain the object itself and it's not a part of the object. In the experimental section we understand that the patch is put in an image (or captured in real settings) and the aim is to detect it as an object. The video provided by the authors illustrate this clearly. But once again, it's not clear while reading the text.

Details about the projection function used are missing.

The authors claim that they could not replicate the digital protocol to real-world settings due to ressource contraints. More explanation should be provided. Is this related to the small number of K patches generated for the real world setting ? Once the data is collected processing it should take as much as in digital settings, no ?

 It's unclear why they had to reduce the angle span in real world settings from -90<->90 to -70<->70? Where the results worse ? The projection function used for training can still reflect real-world transformations when the angle values are beyond 70° ?

What's the impact of choosing the CLIP embedding of <angle-robust> as an initializer ?
The choice of the token <angle-robust> seems quite random as results using the initial representation of the token do not seem conclusive. Choosing other tokens for this purpose is something that you estimate feasible ?

How do you explain the change in behaviour between Table 1 (where NDDA+AngleROCL seems dominant) and Table 1 (in the Supp Mat) where (MAGIC+AngleROCL) seems way much better.

Authors mention that they generate 50 patches for each "remove text or pattern" text prompt and randomly selecting 100 patches as their NDDA baseline. They only mention 100 patches for MAGIC. The overall results only accoung for 100 patches each ? What about the other 50 ? Please explain what -"remove text or pattern" text prompt"- means.

For the physical part, what do you mean by 25 matched pairs of patches from each method.

In section 5.1 authors mention the normalized weight function, but do not define it precisely. Uniform weight ? Gausian centered in 0° ?

The discussion in section 6 (Embedding analysis) is quite unclear. As the Figure 4, is once again to small, the discussion about color/shape and angle robustness is a bit difficult  to follow.

How the approach might work on other target objects ? Does it generalize ?

It would have been interesting to quantify the overhead brought by exploring extensively all the angles. Running time of individual runs for training the "robust_angle" would be interesting to report and compare with equivalent works from the state of the art.

**Ethical Concerns:**

["NO or VERY MINOR ethics concerns only"]

**Final Justification:**

Although the responses where convincing, the original paper requires a lot of improvements in my opinion before publication and I m not certain that the camera ready will be able to fully all concerns raised by the reviewers, even though elements of answers are presented in the rebuttal.

**Limitations:**

The authors mention that they work only considers horizontal angles. But, their approach seems quite generic and in presence of a projection function that can transform the patch considering a 3D angle, 3D angle variations might be addressed. Do the authors forseen others limitations then the extensive training required for more angles ?

**Paper Formatting Concerns:**

Line 60 missing ")"
Figures are all impossible to read.
Tables should use the same fontsize as the main text.

**Quality:**

2

**Strengths And Weaknesses:**

The authors add their contribution on top of state-of-the art generators and largely improve the performances obtained over large angle spans.

The paper is quite unclear on the objectives, the overall methodology and assumes a lot of preknowledge from a generic public as the one in NeurIPS. It took me reading the entire paper to understand that the aim is to generate patches that are not there to deceive the system in a sense where by ading patches to the image, the model stop predicting the expected class. They just imply the term "misleading" (line25) and discuss about "erroneous ouputs" (line 60).

The hyperparameters are set without proper explanation of the experimental settings in which they were set. Only mean pourcentages are provided with no idea on the variance observed through the runs.

Concrete examples that could make things clear are only provided in the supplementary material. Supplementary material should be there to give more insights on specific aspects, but enough elements should be in the main paper in order to let the reviewer fully understand the contribution.

The authors criticize the existing STAR that they only focus on a set of restricted angles. But, they do quite the same, except that they are considering the full range of horizontal angles ranging from -90 to 90 with a step of 1. Would the other train extensively their models, couldn't we expect to obtain the same results ? The authors does not discuss the importance of the angle step considered, nor they try to reduce the angle span to see how much they approach can generalize to angles not seen for training the "robust_angle" tokens.

---

> ### Author Rebuttal · Authors · 2025-07-31
>
> **Important Clarifications on Factual Errors:**
>
> We appreciate your detailed review, but we must first address significant misunderstandings in your summary: (1) **Distillation Model Misconception**: We employ textual inversion, not distillation models. (2) **Training Angle Sampling**: We use 9 fixed angles during training, not -90° to 90° with 1° step (that's our testing protocol in Section 5.1). The response to the main concerns is as follows:
>
> ### 1. **Paper clarity and methodology description**
>
> **Objective:** We generate standalone adversarial patches using T2I models that fool object detectors across multiple viewing angles in digital & physical environments. These patches cause misclassification—making detectors incorrectly identify our generated patch as a legitimate target object (e.g., "stop sign"). **The term `adversarial patch'** refers to a standalone image/object that is generated to be detected as a target object, rather than a modification applied to an existing object.
>
> **Motivation:** Existing T2I patches only work within narrow viewing ranges. Current methods like NDDA succeed at frontal views (0°) but fail at oblique angles (±60°+), limiting real-world applicability (See Fig. 1). Task-specific instructions fail to improve robustness.
>
> **Methodology:** AngleRoCL learns an <angle-robust> token that, when incorporated into prompts like "a photo of a <angle-robust> stop sign," guides diffusion models to generate patches maintaining effectiveness across wide viewing angles.
>
> We will revise the paper for broader accessibility.
>
> ### 2. **Hyperparameter discussion**
>
> Our approach builds upon textual inversion with task-specific hyperparameters: sampled angles and <angle-robust> token initialization. The results in the following two Tables show that 9 sampled angles provide better performance, and "robust" significantly outperforms alternatives by 20+ percentage points due to the alignment with the desired property.
>
> **Table R-4: Ablation study on different numbers of training angles for concept learning against YOLOv5. The AASR is the evaluation metric.**
>
> |Training Angles|Final AASR|Improvement vs. Baseline|
> |:-:|:-:|:-:|
> |1 angle|12.20%|+1.48%|
> |3 angles|20.12%|+9.40%|
> |5 angles|53.85%|+43.12%|
> |7 angles|68.45%|+57.73%|
> |**9 angles** (ours)|**81.82%**|**+71.10%**|
> |11 angles|81.96%|+71.24%|
> |13 angles|80.94%|+70.22%|
>
> **Table R-5: Ablation study on different initialization tokens for <angle-robust>. **
>
> |Initialization Token|Final AASR|Improvement vs. Baseline|
> |:-:|:-:|:-:|
> |**robust** (ours)|**81.82%**|**+71.10%**|
> |stable|61.07%|+50.35%|
> |recognizable|20.59%|+9.87%|
> |perspective|18.88%|+8.16%|
> |angular|17.87%|+7.15%|
> |visible|11.75%|+1.03%|
>
> ### 3. **Physical experiment constraints**
>
> We'd like to clarify the distinction between digital and physical experimental constraints, as the primary bottleneck lies in **data collection**rather than processing.
>
> **Scale Comparison:**
> - Digital: 1,268,505 automated evaluations (1,401 patches × 181 angles × 5 detectors)
> - Physical: Manual printing, positioning, and photographing from multiple angles. Testing 40 patches requires 2 full days. Replicating a digital scale would need ~70 working days plus significant resources.
>
> **Validation Purpose:** Physical experiments demonstrate real-world transferability under authentic conditions (printing artifacts, lighting variations), not statistical equivalence to digital experiments. Our 180 physical patches significantly exceed existing literature validation while remaining feasible.
>
> ### 4 **Generalization to other target objects**
>
> To demonstrate broader applicability, we conducted experiments on two additional objects (fire-hydrant and horse). Table R-3 shows consistent improvements across different object categories. Please refer to the response to the 2nd response to Reviewer KqvV.
>
> ### 5. **Projective transformation function details**
>
> We provide the mathematical description of our projective transformation function Proj(·) used in Eq. (3).
>
> **Mathematical Formulation:**
>
> The function Proj( )  transforms the frontal-view image $\mathbf{I}_{0^\circ}$
>
> to simulate angle $\theta$ observations via homography matrix $\mathbf{H}_\theta$. We have
>
> $\mathbf{I}_\theta (x, y) = \mathbf{I}_0^\circ \left( x',  y' \right)$
>
> where
>
> $$
> x' = \frac{M_{11}^{-1} x + M_{12}^{-1} y + M_{13}^{-1}}{M_{31}^{-1} x + M_{32}^{-1} y + M_{33}^{-1}},
> y' = \frac{M_{21}^{-1} x + M_{22}^{-1} y + M_{23}^{-1}}{M_{31}^{-1} x + M_{32}^{-1} y + M_{33}^{-1}}
> $$
>
> where $M_{ij}$ are elements of $\mathbf{H}_\theta$.
>
> **$\mathbf{H}_\theta$ Calculation Steps:**
> 1. Calculate camera position from angle $\theta$;
> 2. Compute view matrix $\mathbf{V}$ (world-to-camera coordinates);
> 3. Obtain projection matrix $\mathbf{P}$ (camera-to-2D coordinates);
> 4. Combine $\mathbf{V}$ and $\mathbf{P}$ for $\mathbf{H}_\theta$.
>
> This framework maintains differentiability for gradient optimization during AngleRoCL training.
>
> ### 6. **Patch selection in experiment and completed results**
>
> **"Remove Text or Pattern" prompt definition:** These prompts are sourced from the NDDA dataset and involve modifications that remove robust features from stop signs, as detailed in our supplementary materials.
>
> **Patch selection process:** Following NDDA and MAGIC protocols, we generated 50 patches for each of the 8 prompts, totaling 400 patches. We randomly selected 100 patches from this pool for our main experiments.
>
> **Complete Dataset Validation:** We conducted supplementary experiments using the complete 400-patch dataset to address the concern. **Table R-6** demonstrates that the results via our randomly sampled patches accurately represents the complete dataset performance.
>
> **Table R-6: Complete Dataset Validation**
>
> |Environment|Method|Faster R-CNN|YOLOv3|YOLOv5|RT-DETR|YOLOv10|Avg.|
> |:-:|:-:|:-:|:-:|:-:|:-:|:-:|:-:|
> |Environment 1|NDDA|21.60%|1.49%|12.53%|13.11%|8.69%|**11.48%**|
> ||NDDA+AngleRoCL|40.38%|5.19%|35.34%|25.26%|22.56%|**25.75%**|
> |Environment 2|NDDA|36.12%|29.06%|26.71%|41.62%|27.07%|**32.12%**|
> ||NDDA+AngleRoCL|52.48%|48.07%|48.56%|48.19%|44.31%|**48.32%**|
> |Environment 3|NDDA|25.49%|9.11%|23.09%|21.98%|20.48%|**20.03%**|
> ||NDDA+AngleRoCL|44.37%|24.55%|45.11%|33.46%|37.10%|**36.92%**|
> |Environment 4|NDDA|16.94%|3.04%|8.98%|16.01%|11.95%|**11.38%**|
> ||NDDA+AngleRoCL|30.59%|5.94%|25.87%|29.14%|25.85%|**23.48%**|
> |Environment 5|NDDA|33.32%|11.09%|25.28%|48.36%|21.73%|**27.96%**|
> ||NDDA+AngleRoCL|49.27%|24.13%|45.04%|54.34%|37.71%|**42.10%**|
> |Environment 6|NDDA|24.00%|18.02%|18.02%|36.45%|18.99%|**23.10%**|
> ||NDDA+AngleRoCL|41.61%|39.38%|39.38%|45.40%|36.70%|**40.49%**|
>
> ### 7. **Performance difference between Table 1 and Supp Mat Table 1**
>
> Performance differences between digital (Table 1) and physical environments stem from substantial environmental variations.
>
> **Environmental Factors:** Digital uses the nuImage dataset scenes while physical experiments use different geographic locations (different countries), weather conditions, and camera specifications.
>
> **Cross-Validation:** We digitally evaluated patches in physical environment images to verify consistency:  In Table R-7, when physical environments are evaluated digitally, performance ranking (MAGIC+AngleRoCL > NDDA+AngleRoCL) matches physical results, confirming environmental factors drive differences.
>
> **Table R-7: Digital-Physical Environment Cross-Validation. AASR is as the metric**
>
> |Environment|Method|Faster R-CNN|YOLOv3|YOLOv5|RT-DETR|YOLOv10|Avg.|
> |:-:|:-:|:-:|:-:|:-:|:-:|:-:|:-:|
> |Environment 7|NDDA|21.46%|1.56%|21.51%|15.39%|14.55%|**14.89%**|
> ||NDDA+AngleRoCL|28.76%|14.43%|50.16%|33.03%|31.44%|**31.56%**|
> ||MAGIC|14.55%|1.27%|22.35%|5.61%|12.80%|**11.32%**|
> ||MAGIC+AngleRoCL|41.01%|15.61%|45.84%|39.83%|31.62%|**34.78%**|
> |Environment 8|NDDA|24.88%|6.91%|25.56%|23.70%|21.85%|**20.58%**|
> ||NDDA+AngleRoCL|44.71%|13.59%|39.71%|28.48%|31.57%|**31.61%**|
> ||MAGIC|32.97%|6.23%|28.25%|2.52%|20.73%|**18.14%**|
> ||MAGIC+AngleRoCL|52.75%|13.59%|44.38%|39.55%|35.33%|**37.12%**|
> |Environment 9|NDDA|48.48%|12.19%|34.38%|55.16%|24.88%|**35.02%**|
> ||NDDA+AngleRoCL|51.01%|24.66%|53.37%|59.26%|37.30%|**45.12%**|
> ||MAGIC|40.89%|18.48%|42.24%|39.43%|22.97%|**32.80%**|
> ||MAGIC+AngleRoCL|58.03%|43.87%|63.53%|63.25%|42.86%|**54.31%**|
>
> ### 8. **Angle span reduction in real-world settings**
>
> The angle span reduction from ±90° to ±70° was based on two key considerations: efficiency optimization and empirical observations from our digital experiments.
>
> **Justification:** Figure 2 shows most methods have substantially reduced performance beyond ±75°, where geometric visibility constraints dominate over patch design differences. The ±70° range captures meaningful performance variations. Given the significant time and labor costs of physical experiments (as discussed in our previous response), we prioritized the angle ranges where method differences are most pronounced rather than testing angles where all methods uniformly fail.
>
> **Extended Validation:** We tested the full ±90° range to address concerns. The results show that including extreme angles has minimal impact on conclusions, validating our original angle range selection.
>
> **Table R-8: Extended physical validation (-90° to 90°)**
>
> |Method|Faster R-CNN|YOLOv3|YOLOv5|RT-DETR|YOLOv10|Avg.|
> |-|-|-|-|-|-|-|
> |NDDA|31.80%|6.10%|12.00%|44.42%|17.68%|**22.40%**|
> |NDDA+AngleRoCL|55.16%|12.63%|48.00%|55.16%|33.30%|**40.85%**|
> |MAGIC|31.36%|3.37%|13.68%|33.68%|7.58%|**17.93%**|
> |MAGIC+AngleRoCL|66.05%|21.89%|57.26%|70.74%|43.95%|**51.98%**|
>
> ### 9. **Justifications**
>
> (1). "25 matched pairs" means pairs with identical settings except for our <angle-robust> token presence/absence.
> (2). Normalized weight in Sec. 5.1 means uniform weighting: $w(\theta) = \frac{1}{|\Omega|}$ for all angles, meaning equal contribution to AASR calculation.
> (3). Figure 4 concludes that the `<angle-robust>` concept is associated with visual features. We will improve all figures for better readability.

---

> > ### Author Response · Authors · 2025-08-04
> >
> > Dear Reviewer,
> >
> > We appreciate your time in reviewing our manuscript. We submitted our author response on July 30th addressing your concerns.
> >
> > Since the discussion period ends on August 6th, we would like to confirm whether our responses have adequately addressed your issues. If you need further clarification, please let us know.
> >
> > Best regards,
> >
> > Authors

---

> > ### Comment · Reviewer_he3u · 2025-08-04
> >
> > I d like to thanks authors for addressing my concerns, providing detailed responses and running new experiments in such short timelapse.
> >
> > I ll update my scores accordingly.
> >
> > Still, I have the feeling that the paper in its initial form suffers from imprécisions that made the initial understanding difficult. I m not fully convinced that the camera ready version will be able to entirely address all concerns raised and be informative for the reader without complements for authors.

---

> > > ### Author Response · Authors · 2025-08-04
> > >
> > > Dear Reviewer he3u,
> > >
> > > Thank you for your thoughtful review and for acknowledging our efforts to address your concerns. We understand your reservations about the initial paper's clarity and take full responsibility for these communication gaps.
> > >
> > > **Our Commitment:** We will comprehensively revise the paper to address all clarity concerns by enhancing content for a self-contained main paper (including moving critical examples from supplementary material), improving all figures for better readability, and implementing your other valuable suggestions.
> > >
> > > With your detailed feedback as our guide, we are confident we can produce a substantially improved camera-ready version that will be informative and accessible to the broader community. Your constructive criticism has strengthened our work considerably. Thank you.
> > >
> > > Best,
> > >
> > > Authors

---

### Official Review · Reviewer_KqvV · 2025-07-02

**Clarity:** 4
**Significance:** 2
**Originality:** 3
**Rating:** 4
**Confidence:** 4

**Summary:**

The paper addresses an interesting problem about viewpoint-invariant adversarial attacks. NDDA method generates adversarial images that remove distinctive features while misleading object detectors. However, in real-world settings, the attack success rate drops significantly when the image is viewed from non-frontal angles. To address this, the authors propose a method to generate angle-robust adversarial images. Specifically, they introduce a new concept word "angle-robust" trained via textual inversion, and use it during inference. The training leverages feedback from YOLOv5. For physical deployment, the generated images can be printed and attached to the target object using a standard printer.

**Questions:**

- In the left side of Figure 4 (ablation study), what exactly does “direct optimization” refer to in terms of training setup?
- Was there a specific reason for choosing textual inversion over other fine-tuning methods such as LoRA?

**Ethical Concerns:**

["NO or VERY MINOR ethics concerns only"]

**Final Justification:**

The problem posed is interesting and it has been addressed well. However, the reviewer still find the scope of the problem to be somewhat narrow (although the reviewer thanks for the author responses which read carefully), so I will maintain my original rating.

**Limitations:**

yes; The authors well addressed their limitations.

**Paper Formatting Concerns:**

No. There is no formatting issue for me.

**Quality:**

4

**Strengths And Weaknesses:**

Strengths
- The paper points out a novel and overlooked problem and provides experimental evidence that NDDA attacks are less effective from non-frontal viewpoints.
- The idea of learning an angle-robust concept using textual inversion without requiring separate training for each angle is novel and interesting.
- The method shows consistent improvement across multiple detectors. The authors also provide extensive analysis to support and clarify the approach.

Weaknesses
- While the reviewer agrees that the identified problem is novel, it is unclear how critical it is from a safety standpoint. For example, in the case of stop signs, only frontal views are typically relevant for driving decisions, so the practical risk may be limited.
- The proposed method is evaluated only on stop signs. It is unclear whether the same angle-dependent issue arises for other objects. In contrast, NDDA includes experiments on a broader set of objects.

---

> ### Author Rebuttal · Authors · 2025-07-31
>
> ### 1. Practical safety significance of multi-angle attacks
>
> **Response**:
>
> Thank you for this thoughtful concern about practical safety implications. While frontal views are indeed primary for driving decisions, angle robustness represents a significant real-world security challenge supported by both practical scenarios and extensive research attention.
>
> **Angle Robustness is Safety-Critical in the Real World:**
>
> - **Traffic signs at intersections are viewed from various angles as vehicles approach from different directions.** For example, in the highway merging and lane changes, drivers observe traffic signs at oblique angles when merging or changing lanes around construction zones and temporary traffic control.
>
> - **Autonomous vehicle sensor arrays**: Modern AVs like Tesla, Waymo use 8+ cameras covering 360° views, continuously processing visual information from multiple perspectives simultaneously—angle-robust patches could compromise their entire perception system.
>
> - **Even "frontal" approaches involve angle variations due to road curvature, vehicle positioning, and sensor placement.** In particular, for the **emergency scenarios**: First responders and emergency vehicles often approach intersections from non-standard angles where traditional patches would fail.
>
> **Extensive Research Validation:** The critical importance of angle-robust adversarial attacks has been acknowledged by substantial research investment: multi-view adversarial examples [2, 12], 3D adversarial patches [7, 21], viewpoint-invariant adversarial noise [35]. This sustained research focus across premier venues (CVPR, ICCV, USENIX Security, NeurIPS) demonstrates that the research community recognizes angle robustness as a fundamental security challenge requiring urgent attention, rather than merely a theoretical concern.
>
> ### 2. Attack results on other objects
>
> **Response:**
>
> Thank you for the insightful question. To address the concern, we conducted additional experiments on two other target objects to demonstrate the broader applicability of our method.
>
> **Experimental Design:** Following NDDA's evaluation scope, we selected fire-hydrant and horse as additional target objects to evaluate object-specific training capabilities. These objects represent different categories—urban infrastructure (fire-hydrant) and natural subjects (horse)—allowing us to assess our method's versatility across diverse visual domains.
>
> **Object-Specific Training Experiment:** We trained dedicated angle-robust concepts for the two target objects, respectively, using their respective NDDA prompt templates. Then, we embed the learned angle-robust concepts into the NDDA prompt to generate angle-robust adversarial patches. **Table R-3** presents the comparative results with/without AngleRoCL:
>
> **Table R-3: Object-Specific Angle-Robust Concept Evaluation**
>
> |Target Object|Method|Faster R-CNN|YOLOv3|YOLOv5|RT-DETR|YOLOv10|Avg.|
> |:-:|:-:|:-:|:-:|:-:|:-:|:-:|:-:|
> |Fire-hydrant|NDDA|16.78%|3.63%|5.50%|23.39%|11.51%|**12.16%**|
> ||NDDA+AngleRoCL|23.95%|6.62%|12.89%|27.14%|19.84%|**18.09%**|
> |Horse|NDDA|19.91%|12.59%|10.32%|26.74%|13.58%|**16.63%**|
> ||NDDA+AngleRoCL|31.86%|23.34%|21.94%|44.70%|24.64%|**29.30%**|
>
> _Note: All experiments in Table R-3 were conducted in Digital Environment 1. For each NDDA prompt, we generated 5 patches, and the reported results represent the average performance across all generated patches._
>
> The results demonstrate that our framework successfully generates effective angle-robust concepts across different target objects, achieving consistent improvements in all categories. This validates that the angle robustness challenge is not unique to stop signs but represents a broader limitation in existing T2I adversarial patch generation methods.
>
> ### 3. "Direct Optimization" definition in ablation study
>
> **Response**:
>
> Thank you for your comments. "Direct optimization" refers to a baseline approach like [35] where we directly optimize adversarial patch pixels using backpropagation to maximize detection confidence across multiple viewing angles, rather than learning a reusable text embedding concept. To further address the concerns, here are the setup details of the baseline:
>
> **Training Setup:**
>
> - **Optimization Target**: Patch pixel values (3×512×512 tensor) vs. AngleRoCL's text embeddings
> - **Loss Function**: Same loss function as AngleRoCL
> - **Angles**: Same training angles as AngleRoCL
> - **Detector**: Same YOLOv5 feedback as AngleRoCL
> - **Optimizer**: Adam optimizer vs. AngleRoCL's AdamW on embeddings
>
> ### 4. Choice of textual inversion instead of LoRA
>
> **Response**:
>
> Thank you for this insightful question. We chose textual inversion over LoRA for several principled reasons related to our method's core objectives and practical considerations.
>
> **1. Plug-and-Play Capability:** **Textual Inversion** enables seamless integration without affecting base model functionality - users simply load our trained embedding and modify prompts from "a blue square stop sign" to "a `<angle-robust>` blue square stop sign" to activate angle robustness. **LoRA**, despite using low-rank adaptation theory (decomposing weight updates as ΔW = BA where B∈R^(d×r), A∈R^(r×k), r<<d), modifies UNet layers and affects all generations even when angle robustness is not desired, compromising the model's normal text-to-image capabilities.
>
> **2. Concept Interpretability:** **Textual Inversion** enables direct semantic analysis of learned robustness concepts through embedding analysis, as demonstrated in our discussion (Line 329 - 349) where we show the learned concept develops meaningful associations with color and shape features, providing insights into which visual attributes contribute to angle robustness. **LoRA** distributes learned features across multiple transformer layers, making it significantly more challenging to isolate and interpret the specific concept being learned.
>
> **3. Training Stability and Parameter Efficiency:** **Textual Inversion** only trains the `<angle-robust>` embedding approximately 768 parameters while preserving all other embeddings and network parameters. **LoRA** trains more parameters (797,184 parameters, 0.09% of UNet with rank=4). However, we observed poor fitting performance that led to severe concept drift during our preliminary experiments - the learned representations progressively deviated from stop sign generation, resulting in images that could barely be detected as stop signs, severely compromising generation quality. We believe a novel learning strategy should be studied for the LoRA for angle-robust generation.
>
> Our approach prioritizes learning transferable, interpretable concepts that enhance rather than modify the base model's capabilities, making textual inversion the better choice for our angle robustness objective.

---

> ### Comment · Reviewer_KqvV · 2025-08-06
>
> Thank you for providing author rebuttal. The reviewer thinks the paper is enough to be accepted and solve interesting problem. The authors also provided additional experiments that the reviewer requested and it works well. However, despite of the rebuttal about practical significance of multi-angle attacks, the reviewer still thinks the problem is only useful for specific situation (e.g. intersection and road curvature). Therefore, the reviewer maintains the original rating.

---

> > ### Author Response · Authors · 2025-08-06
> >
> > Dear Reviewer KqvV,
> >
> > Thank you for your constructive feedback and for acknowledging that our paper addresses an interesting problem. We appreciate your thorough review throughout the process.
> >
> > We understand your perspective on the practical significance. In our revised version, we will explore and validate our method across more diverse application scenarios to better demonstrate the broader real-world relevance of angle-robust adversarial attacks.
> >
> > We greatly appreciate your helpful suggestions during this process.
> >
> > Best regards,
> >
> > Authors

---

### Official Review · Reviewer_T7Kc · 2025-07-05

**Clarity:** 3
**Significance:** 3
**Originality:** 3
**Rating:** 5
**Confidence:** 3

**Summary:**

This paper identifies a key limitation of current Text-to-Image (T2I) adversarial attacks that their effectiveness drops sharply when viewed from different angles. The authors propose Angle-Robust Concept Learning (AngleRoCL), a plug-and-play approach that learns a textual embedding  "angle-robust" representing angle robustness, which can be inserted into any T2I prompt to guide the generation of patches that maintain attack efficacy across a wide range of viewing angles. Experiments show that the proposed approach achieves the strong attack success rate of 65.86% (Table 2), significantly outperforming baseline methods.

**Questions:**

- Since your concept embedding "angle-robust" is trained with YOLOv5's feedback, have you attempted training with multiple detectors jointly or rotating detectors during training? This might reduce potential detector-specific bias and improve generalization. Did you explore this possibility?
 - Recent works like DiffPatch [1] and Diffusion to Confusion (2023) also use diffusion models to generate physically robust adversarial patches. Could you clarify how your method compares in terms of real-world robustness and patch realism.

**Ethical Concerns:**

["NO or VERY MINOR ethics concerns only"]

**Final Justification:**

The authors have alleviated all of the reviewer's concerns. Hence, the reviewer will maintain the rating.

**Limitations:**

yes

**Paper Formatting Concerns:**

None that the reviewer could observe.

**Quality:**

3

**Strengths And Weaknesses:**

**Strengths:**
 - The reviewer likes the idea of AngleRoCL’s “angle” robustness for physical-world attacks proposed in the paper as most methods generally focus on realism without any concrete definition. Further, AngleRoCL’s textual embedding-based approach is compatible with any prompt and attack pipeline (e.g., MAGIC, NDDA), and is model-agnostic.
 - Experiment-wise, the authors conducts large-scale experiments across digital and real-world physical environments, including deployment via printed patches, on five detectors (YOLOv3, v5, v10, Faster R-CNN, RT-DETR), using a new angle-aware metric (AASR), which integrates attack success over continuous viewing angles. For example, MAGIC+AngleRoCL improves physical-world AASR from 22.72% to 65.86%.

----
**Weaknesses:**
 - AngleRoCL uses YOLOv5 alone for training supervision, which biases the learned embedding toward this detector’s feature sensitivities. Although generalization to other detectors is tested, improvements are uneven, e.g., larger for YOLOv5 than for RT-DETR. This compromises the method’s claim of full model-agnosticism.
 - While it is novel in angle robustness, the paper does not compare directly to methods like DiffPatch [1] or Diffusion to Confusion [2], which also use diffusion models for generating robust patches in physical environments. Even if their objectives differ (stealth vs. view-invariance), a head-to-head comparison would better position AngleRoCL and strengthen claims of superiority.

----
[1] DiffPatch: Generating Customizable Adversarial Patches using Diffusion Models, arXiv 2024 \
[2] Diffusion to Confusion: Naturalistic Adversarial Patch Generation Based on Diffusion Model for Object Detector, arXiv 2023

---

> ### Author Rebuttal · Authors · 2025-07-31
>
> ### 1. YOLOv5 Training Bias, Model-Agnosticism Claims, and Multi-Detector Training
>
> **Response**:
>
> Thank you for this insightful question about detector-specific bias. We appreciate your observation and would like to clarify our claims and provide a detailed analysis.
>
> **Clarification of Model-Agnosticism Claims:** Our original claim about the "model-agnostic" referred to the ability to achieve consistent angle robustness improvements across unseen, black-box detectors during testing, rather than uniform improvement magnitudes across all architectures. The terminology may be misleading, and we will revise our manuscript to provide a clearer characterization of our approach's capabilities and limitations. While we use YOLOv5 to guide concept learning during training, we evaluate on four additional detectors (Faster R-CNN, YOLOv3, RT-DETR, and YOLOv10) that provide no guidance information during the generation process. As demonstrated in Table 1 in the submission, AngleRoCL enhances performance across all tested detectors in six environments, demonstrating cross-detector transferability.
>
> **Regarding Multi-Detector Training:** Intuitively, training with multiple detectors should enable learning a concept that achieves robust enhancement across different detector architectures. However, our implementation revealed that multi-detector training using both YOLOv5 and RT-DETR simultaneously encounters significant optimization challenges, with the loss function failing to converge effectively. As demonstrated in Table R-1, our single-detector training approach achieves substantially higher Angle-Aware Attack Success Rates (AASR) compared to the multi-detector variant, suggesting that naively combining multiple detector objectives creates conflicting optimization signals. These findings indicate that developing effective multi-detector training strategies represents an important avenue for future research, as the challenge likely stems from fundamental differences in how various detector architectures process visual features. While our current single-detector approach demonstrates strong cross-detector generalization, developing truly detector-agnostic angle-robust concepts remains an open challenge.
>
> **Table R-1: Multi-detector training vs single-detector training across detectors**
>
> |Training Strategy|Method|Faster R-CNN|YOLOv3|YOLOv5|RT-DETR|YOLOv10|Avg.|
> |:-:|:-:|:-:|:-:|:-:|:-:|:-:|:-:|
> |Single-Detector Training|NDDA|34.34%|11.82%|11.45%|25.16%|18.20%|**20.19%**|
> ||NDDA+AngleRoCL|54.12%|33.49%|47.17%|44.87%|43.05%|**44.54%**|
> |Multi-Detector Training|NDDA|34.34%|11.82%|11.45%|25.16%|18.20%|**20.19%**|
> ||NDDA+AngleRoCL|25.67%|8.45%|2.21%|12.15%|4.41%|**10.58%**|
>
> _Note: All experiments in Table R-1 were conducted in Digital Environment 1. For each NDDA prompt, we generated 5 patches using the trained robust concept, and the reported results represent the average performance across all generated patches._
>
> ### 2. Comparisons with Recent Diffusion-based Methods
>
> **Response:**
>
> Thank you for providing two related works. However, **DiffPatch and Diffusion to Confusion (D2C) address fundamentally different research objectives compared to our method.** DiffPatch and D2C focus on generating adversarial patches for human evasion (e.g., T-shirt patches to avoid person detection), while our work specifically targets angle robustness for object detection attacks—a previously unexplored dimension in T2I adversarial patch generation.
>
> **Our Comparison Strategy:** While direct comparison is challenging due to divergent objectives, we evaluate these methods on angle robustness using our proposed AASR metric to demonstrate the critical gap in existing T2I adversarial patch methods.
>
> **Implementation and Initial Results:** We contacted both research teams for code access. The DiffPatch authors graciously provided their implementation, enabling us to reproduce their results and evaluate the generated patches for angle robustness under our experimental protocol. Unfortunately, the D2C authors have not responded and no public code is available, limiting our comparison scope to DiffPatch.
>
> **Table R-2: Angle Robustness Comparison with DiffPatch**
>
> |Method|Faster R-CNN|YOLOv3|YOLOv5|RT-DETR|YOLOv10|Avg.|
> |:-:|:-:|:-:|:-:|:-:|:-:|:-:|
> |DiffPatch|0.00%|11.89%|33.81%|0.00%|1.88%|**9.52%**|
> |NDDA+AngleRoCL|54.12%|33.49%|47.17%|44.87%|43.05%|**44.54%**|
>
> _Note: The experiments in Table R-2 were conducted using patches trained on the INRIA person dataset with YOLOv5 as the target detector. We applied horizontal projective transformations same as our AngleRoCL to the patches and evaluated them against test set images as backgrounds to assess angle robustness performance._
>
> Our results in Table R-2 demonstrate that DiffPatch exhibits significant angle robustness limitations when evaluated across multiple viewing angles—performing similarly to baseline methods like NDDA that lack angle-aware optimization. **This comparison validates our core contribution: existing T2I adversarial patch methods, regardless of their sophistication in other dimensions, fail to address the critical challenge of maintaining attack effectiveness across diverse viewing angles in physical deployments.**
>
> **Future Work Integration:** The complementary nature of these approaches presents exciting opportunities for future research. Our angle-robust concept learning could be integrated with DiffPatch's customization framework to create patches that are simultaneously natural-looking, customizable, and angle-robust. Additionally, comprehensive evaluation including patch realism assessments using user preference studies and naturalness metrics would provide a more complete comparison framework. We will add the discussion into our revision and cite the two works.

---

> > ### Comment · Reviewer_T7Kc · 2025-08-03
> >
> > Hi Authors,
> >
> > Thank you for your hard work on the rebuttal. Your responses have alleviated all of the reviewer's concerns. Hence, the reviewer will maintain the rating. Best of luck!

---

> > > ### Author Response · Authors · 2025-08-04
> > >
> > > Dear Reviewer,
> > >
> > > Thank you for your constructive feedback and for taking the time to review our rebuttal. We appreciate your positive assessment and are grateful for your support.
> > >
> > > Best regards,
> > >
> > > Authors

---

### Decision · Program_Chairs · 2025-09-17

**Decision:**

Accept (poster)

**Comment:**

This paper addresses an interesting problem about viewpoint-invariant adversarial attacks using text-to-Image (T2I) generated patches. The authors propose Angle-Robust Concept Learning (AngleRoCL), a plug-and-play approach that learns a textual embedding "angle-robust" representing angle robustness, which can be inserted into any T2I prompt to guide the generation of patches that maintain attack efficacy across a wide range of viewing angles. The training leverages feedback from YOLOv5. For physical deployment, the generated images can be printed and attached to the target object using a standard printer.

The submitted paper shows results only on STOP signs. However, in rebut, the authors provided results on additional objects though the range is still very limited. The authors also clarified that the method uses YOLOv5 during training but the attack also applies to other detectors. Overall, the reviewers have a more positive view of the papers after rebut discussions but there is concern if those additional clarifications and experiments will be incorporated in the final version. There also lingering concerns about the generality and applicability of the method.

The ACs recommend that the paper be accepted but the authors need to take care that the revised version includes the key elements from the rebut discussion.